# Defibrotide impact on the acute GVHD disease incidence in pediatric hematopoietic stem cell transplant recipients

Domenica Squillaci[1] , Annalisa Marcuzzi[2], Erika Rimondi[3] , Guglielmo Riccio[1], Egidio Barbi[1,4], Davide Zanon[4], Natalia Maximova[4]

Despite advances in acute graft-versus-host disease (aGVHD) prophylaxis, current pharmacological approaches fail to prevent aGVHD. The protective effect of defibrotide on GVHD incidence and GVHD-free survival has not been sufficiently studied. 91 pediatric patients included in this retrospective study were divided into two groups based on defibrotide use. We compared the incidence of aGVHD and chronic GVHD-free survival between the defibrotide and control groups. The incidence and severity of aGVHD were significantly lower in patients who received defibrotide prophylactic administration than in the control group. This improvement was observed in the liver and intestinal aGVHD. No defibrotide prophylaxis benefit was observed in the prevention of chronic GVHD. The pro-inflammatory cytokine levels were significantly higher in the control group. Our findings suggest that prophylactic administration of defibrotide in pediatric patients significantly reduces the incidence and severity of aGVHD, with a modification of cytokine pattern, both strongly coherent with the protective drug's action. This evidence adds to pediatric retrospective studies and preclinical data suggesting a possible defibrotide role in this setting.

## Introduction

Allogeneic hematopoietic stem cell transplantation (allo-HCT) is a highly specialized medical procedure that introduced the first regenerative approach to clinical practice nearly 60 yr ago (1, 2).

Acute graft-versus-host disease (aGVHD) is a common life-threatening complication of allogeneic HSCT, distinguished by systemic inflammation mostly attacking the liver, skin, and gut, which occurs in 25% to 50% of patients. Acute GVHD is the second most common cause of death in allo-HCT recipients after relapse of the primary disease (3). Despite aGVHD frequency having decreased over time in matched related and unrelated donor transplantations, the number of patients experiencing this complication has increased because of the growing number of allo-HCT performed worldwide (4). Even with advances in GVHD prophylaxis, current pharmacological approaches fail to prevent aGVHD effectively, implicating an essential medical need for developing novel therapies (5, 6, 7).

Increasing evidence suggests that angiogenesis and endothelial injury are primarily involved in GVHD (8, 9, 10, 11). Furthermore, endothelial damage correlates with the risk of steroid refractoriness and mortality in patients with severe GVHD (12, 13).

The endothelium is the first contact for blood immunological effector cells and plays a regulatory role in various inflammatory processes. Substantial scientific evidence suggests some early transplant-related complications, such as sinusoidal obstruction syndrome (SOS), capillary leak syndrome, vascular idiopathic pneumonia syndromes, engraftment syndrome, transplant-associated thrombotic microangiopathies, originate from localized or systemic endothelial cell damage (14). The common origin shared between vascular endothelial syndromes and aGVHD had already been speculated several years ago. It has been considered that angiogenesis might play a significant role in aGVHD development. There are enough evidences that angiogenesis in GVHD occurs exclusively in target organs and precedes leukocyte infiltration during GVHD (15), occurring as early as day +2 after HSCT. Immune cell infiltration is known to contribute to resistance against antiangiogenic drugs. Thus, early therapy onset could diminish these mechanisms and inflammatory-associated side effects. In the setting of GVHD, early treatment leads to preventing or minimizing disease outbreak (16).

Defibrotide is a 90% single-stranded and 10% double-stranded polydisperse blend of phosphodiester oligonucleotides derived from controlled depolymerization of porcine intestinal mucosal DNA. It has been described as a multi-target drug, showing antithrombotic/thrombolytic, anti-ischemic, pro-fibrinolytic, and antiangiogenic effects (17, 18, 19). Moreover, defibrotide has also shown important anti-inflammatory properties and a protective effect on endothelial cells from HSCT conditioning (20, 21). It has proven effective for the prophylaxis and treatment of SOS, another life-threatening complication of endothelial origin after allo-HCT.

---

[1]Department of Medical, Surgical and Health Sciences, University of Trieste, Trieste, Italy  [2]Department of Translational Medicine, University of Ferrara, Ferrara, Italy  [3]Department of Translational Medicine and LTTA Centre, University of Ferrara, Ferrara, Italy  [4]Department of Pediatrics, Institute for Maternal and Child Health - IRCCS Burlo Garofolo, Trieste, Italy

Correspondence: natalia.maximova@burlo.trieste.it

Several clinical studies have examined whether defibrotide can reduce the incidence of aGVHD ([22], [23], [24], [25], [26], [27], [28], [29], [30]). However, the literature on randomized defibrotide assessments for GVHD prevention is relatively tiny and reports disagreeing conclusions.

We retrospectively analyzed the population of pediatric patients undergoing an allo-HSCT in our center in the last decade to assess the impact of defibrotide on the incidence and severity of aGVHD.

# Results

This study included 91 patients (57 males and 34 females) divided among the defibrotide group (47) and the control (44) group. The median age at HSCT of the entire cohort was 9.33 yr (5.18–13.30), whereas in the defibrotide and control groups was 10.89 yr (5.48–13.83) and 8.11 yr (4.76–11.52), respectively. Underlying diagnoses were grouped into acute leukemia (65%), myelodysplastic syndrome (16%), solid tumor (4%), and nonmalignant disease (14%). The most frequent diagnosis within the acute leukemia subgroup was acute lymphoblastic leukemia (63%), whereas the most frequent diagnosis in the nonmalignant subgroup was thalassemia major (54%). Statistically significant differences in the baseline patient characteristics were not found, except in the year of transplant, more latest in the control group. Detailed patient demographics are shown in Table 1.

Defibrotide administration was started on the first day of conditioning and terminated on day +28, regardless of engraftment, for all patients undergoing prophylaxis. The mean duration of defibrotide prophylaxis in the defibrotide group was 34.9 d (range 34–36 d).

In the Kaplan-Meier survival analysis, the cumulative survival rate showed no statistically significant differences ($P > 0.05$) between the two groups (Fig 1A). We compared the incidence of aGVHD in the defibrotide and control groups. The overall incidence of aGVHD in the defibrotide group was 23% (11 patients). In the control group, the overall incidence was 50% (22 patients) therefore, significantly higher ($P = 0.010$). The differences were particularly marked when comparing the severity of aGVHD. The incidence of aGVHD grade II–IV in the defibrotide group was 4% (2 patients) versus 39% (17 patients) in the control group ($P = 0.001$). The aGVHD-free survival was significantly higher ($P = 0.047$) in the defibrotide group as compared with the control group (Fig 1B). We generated a forest plot to illustrate the protective effect of defibrotide on the development and severity of aGVHD (Fig 2). In terms of the overall impact, the odds ratio (OR) of developing any grade of aGVHD in defibrotide prophylaxis was 0.306 (95% confidence interval [CI] 0.125–0.750). Furthermore, the OR of developing moderate to severe aGVHD was 0.108 (95% CI 0.029–0.404). Investigating the incidence of aGVHD stratified by organs involved showed the protective effect of defibrotide on the liver (OR 0.163 [95% CI, 0.049–0.538], $P = 0.003$) and intestinal forms of GVHD (OR 0.106 [95% CI, 0.022–0.503], $P = 0.005$), but not on the cutaneous variant (OR 0.440 [95% CI, 0.163–1.184], $P = 0.104$). We used multivariate logistic regression analysis to estimate the independent association between aGVHD and defibrotide prophylaxis (Fig 3). Logistic regression showed that

the protective effect of defibrotide prophylaxis on aGVHD is maintained after adjustment for confounding factors (OR 0.350; 95% CI, 0.136–0.899).

We compared the log-rank test's chronic GVHD-free survival between the two groups (Fig S1). We did not find a statistically significant difference ($P = 0.749$).

We compared 27 pro-inflammatory and anti-inflammatory cytokine blood levels in the defibrotide group with the same cytokine cluster in the control group (Table 2). We found statistically significant differences in most analyzed cytokines except for IL-1$\beta$, IL-12, GM-CSF, and VEGF. The levels of the pro-inflammatory cytokines, such as IL-7, IL-6, IL8, IP-10, MCP-1, MIP-1$\alpha$, MIP-1$\beta$, TNF-$\alpha$, and RANTES, were significantly higher in the control group.

An inverse association was found for the anti-inflammatory cytokines, such as IL-1ra and IL-17, with significantly higher values in the defibrotide group. The case of SOS in the defibrotide group was not documented. On the other hand, 13.6% (6 patients) of proven SOS were found in the control group ($P < 0.05$).

# Discussion

The use of defibrotide in the treatment of SOS is widely described. Some preclinical studies have analyzed the impact of defibrotide on the incidence of GVHD. García-Bernal et al used a mouse model of GVHD after allo-HCT, demonstrating that defibrotide, in both prophylaxis and treatment, effectively prevents T cell and neutrophil infiltration and tissue damage associated with GVHD, thereby reducing the incidence and severity of aGVHD. In vitro studies on human cells revealed that defibrotide inhibits leukocyte–endothelial interactions by down-regulating the expression of key endothelial adhesion molecules involved in leukocyte trafficking ([33]).

Previously published studies on adults who received defibrotide prophylaxis for SOS provided inconsistent data on the defibrotide's ability to reduce the incidence of aGVHD. Akpinar et al studied 38 adult patients undergoing HSCT who received defibrotide in prophylaxis to prevent SOS. The cumulative incidence of acute grade III–IV GVHD and moderate/severe chronic GVHD requiring 1-yr systemic immunosuppression was 20.6% and 5.3%, respectively. Relapse-free mortality, GVHD relapse-free survival, and overall survival in the 1-yr study cohort were 21.1%, 44.7%, and 57.9%, respectively ([34]). Strouse et al found notable differences in the cumulative incidence of grade II–IV acute GVHD at day 100 post-HCT in patients who received defibrotide versus those who did not receive defibrotide (23.1% versus 37.7%; difference, −14.6 [95% CI: −33.1, 3.9]) ([27]). Chalandon et al indicated that defibrotide prophylaxis significantly reduced the 1-yr cumulative incidence of aGVHD ([28]). Tekgündüz's study of 195 HSCT recipients showed that the incidence of acute GVHD was 26% for patients who received defibrotide before HSCT, 40% for those who received defibrotide after HSCT, and 47% for those who received no defibrotide ($P = 0.057$), with a trend toward a lower rate of severe GVHD in the pre-HSCT arm than in the other groups ($P = 0.051$) ([24]). A recent study reports that defibrotide prophylaxis may benefit the current standard of care to prevent

**Table 1.  Patient demographics.**

| Baseline characteristics | Defibrotide group (n = 47) | Control group (n = 44) | *P*-value |
|---|---|---|---|
| Age at transplant, years, median (IQR) | 10.89 (5.48–13.83) | 8.11 (4.76–11.52) | 0.726 |
| Sex, number (%): | | | 0.808 |
| Male | 30 (64) | 27 (61) | |
| Female | 17 (36) | 17 (39) | |
| Primary diagnosis, number (%): | | | 0.529 |
| Acute leukemia | 30 (64) | 29 (66) | |
| Myelodysplastic syndrome | 9 (19) | 6 (14) | |
| Solid tumor | 3 (6) | 1 (2) | |
| Nonmalignant disease | 5 (11) | 8 (18) | |
| Disease stage at HSCTa, number (%): | | | 0.330 |
| Early | 14 (30) | 20 (45) | |
| Intermediate | 12 (26) | 8 (18) | |
| Late | 10 (21) | 5 (11) | |
| Untreated | 11 (23) | 11 (25) | |
| Recipient CMV serostatus, number (%): | | | 0.397 |
| Positive | 35 (74) | 36 (82) | |
| Negative | 12 (26) | 8 (18) | |
| Donor type, number (%): | | | 0.177 |
| Sibling | 19 (40) | 11 (25) | |
| MUD | 23 (49) | 30 (68) | |
| Haploidentical | 5 (11) | 3 (7) | |
| Donor-recipient sex-matched, number (%): | | | 0.392 |
| Matched | 21 (45) | 21 (48) | |
| Male/female mismatched | 14 (30) | 8 (18) | |
| Female/male mismatched | 12 (26) | 15 (34) | |
| Type of conditioning, number (%): | | | 0.151 |
| TBI-based | 23 (49) | 15 (34) | |
| Busulfan-based | 24 (51) | 29 (66) | |
| Graft source, number (%): | | | 0.007 |
| Bone marrow | 38 (81) | 24 (55) | |
| Peripheral stem cells | 9 (19) | 20 (45) | |
| GVHD prophylaxis, number (%): | | | 0.177 |
| Tacrolimus | 19 (40) | 11 (25) | |
| Tacrolimus + MMF | 23 (49) | 30 (68) | |
| Tacrolimus + MMF + PTCy | 5 (11) | 3 (7) | |
| ATG used, number (%) | 23 (48) | 15 (34) | 0.151 |

[a]Classified according to Gratwohl for hematological malignancies and Thakar et al for nonmalignant disease (31, 32).
IQR, interquartile range; HSCT, hematopoietic stem cell transplantation; CMV, cytomegalovirus; MUD, matched unrelated donor; TBI, total body irradiation; MMF, mycophenolate mofetil; PTCy, posttransplant cyclophosphamide; ATG, anti-thymocyte globulin.

aGVHD without significant toxicities. However, observed differences in endpoints between the two arms were not substantial (29). In contrast, a recent retrospective study by Tilmont et al showed no protective effect of defibrotide on the development or severity of aGVHD (30). Similar results were also obtained in the phase 2 open-

label trial, completed in May 2020 (NCT03339297), evaluating defibrotide to prevent aGVHD after HCT in children and adults (35).

In the pediatric field, concordant results were obtained. In the phase 3 VOD/SOS prevention study, Corbacioglu et al demonstrated that the 177 transplant patients who received defibrotide

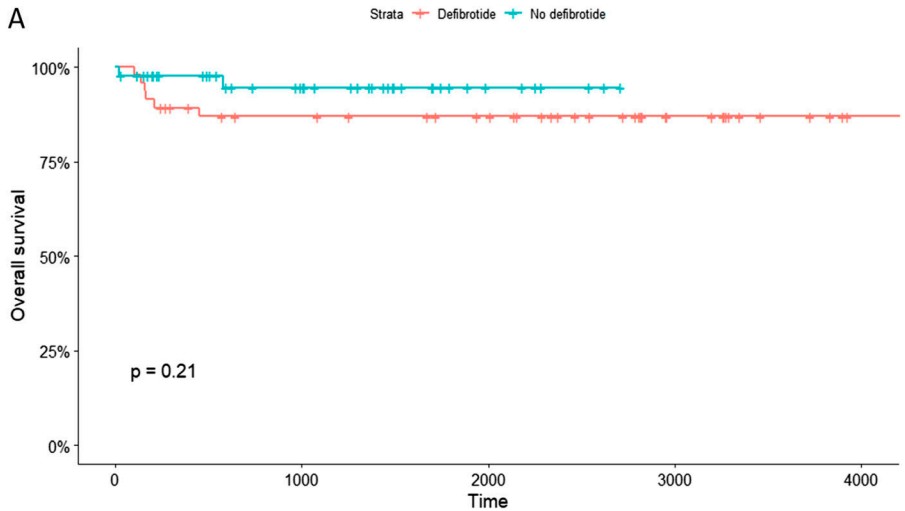

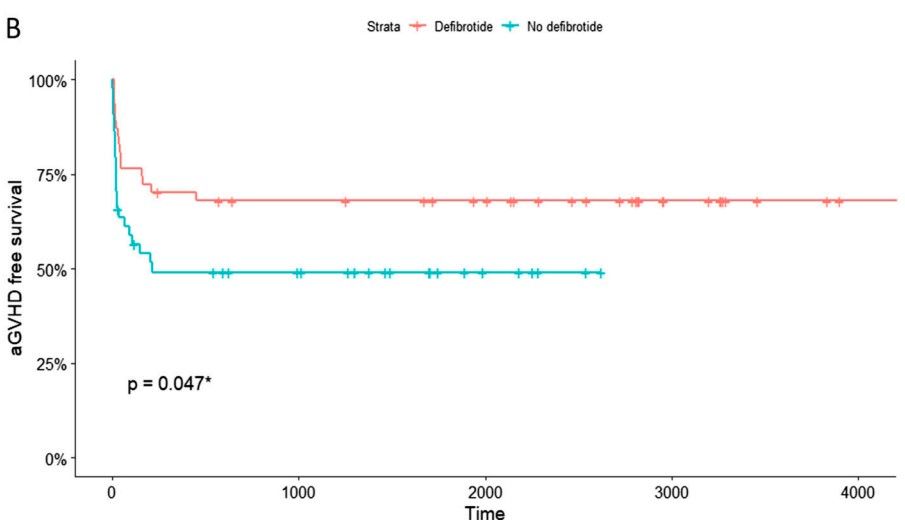

**Figure 1.  Overall survival and acute GVHD-free survival curves.**
**(A)** Kaplan-Meier overall survival curves show no statistically significant differences between the defibrotide and control groups (*P* = 0.21). **(B)** Acute GVHD-free survival is significantly higher in the defibrotide group compared with the control group (*P* = 0.047). aGVHD, acute graft-versus-host disease.

prophylaxis for SOS had a significantly lower incidence and severity of GVHD at 30 and 100 d compared with those in the control group (22). In another randomized phase II pediatric trial of defibrotide in SOS, the incidence and the severity of aGVHD at days +30 and +100 were significantly lower in the defibrotide-treated arm in HSCT recipients (23).

We could speculate that the differences between pediatric and adult outcomes in defibrotide prophylaxis of aGVHD derive from better thymic function in children and significant differences in specific cytokines, B cell, and Treg populations between children and adults (36). Furthermore, these differences may reflect the dissimilarities of immune reconstitution between adult and pediatric transplant recipients and the heterogeneity of underlying diseases, particularly pediatric malignant and nonmalignant diseases (37, 38).

Our findings suggest that prophylactic administration of defibrotide (25 mg/kg daily dose) started on the first day of conditioning and went up to day +28, which significantly reduced the incidence and severity of aGVHD. We observed this advantage in the liver and intestinal aGVHD, but not cutaneous involvement. There was not

any benefit observed in preventing chronic GVHD (cGVHD). Hemorrhagic events of iatrogenic origin or SOS have not been recorded in patients treated with defibrotide. In the safety analysis of the drug, observing the onset of early (infections, rejection, and radio-chemo toxicity) and late (immunological and endocrinological insufficiency) complications, there were no important differences between the two groups. Cytokine profile analysis showed a set of down-regulated (IL-7, IL-6, IL8, IP-10, MCP-1, MIP-1a, MIP-1b, TNF-α, and RANTES) and up-modulated (IL-1ra, VEGF, and IL-17) cytokines after treatment with defibrotide (Fig 4). The values of evaluated cytokines are shown in Table 2.

Our study demonstrated a decrease in IL-6, IL8, and MCP-1 compared with the control group. This observation aligns with previous studies that presented a predominant role of these cytokines in initiating the inflammatory platform and GVHD (39, 40, 41). Of note, the significant increase of IL-8, IL-6, MIP-1α, MIP-1β, and TNF-α was observed in allo-HCT patients and was correlated with a decrease in overall survival (42).

The crucial role of high plasma levels of IL-7 in the incidence of aGVHD and cGVHD is supported by many preclinical and clinical

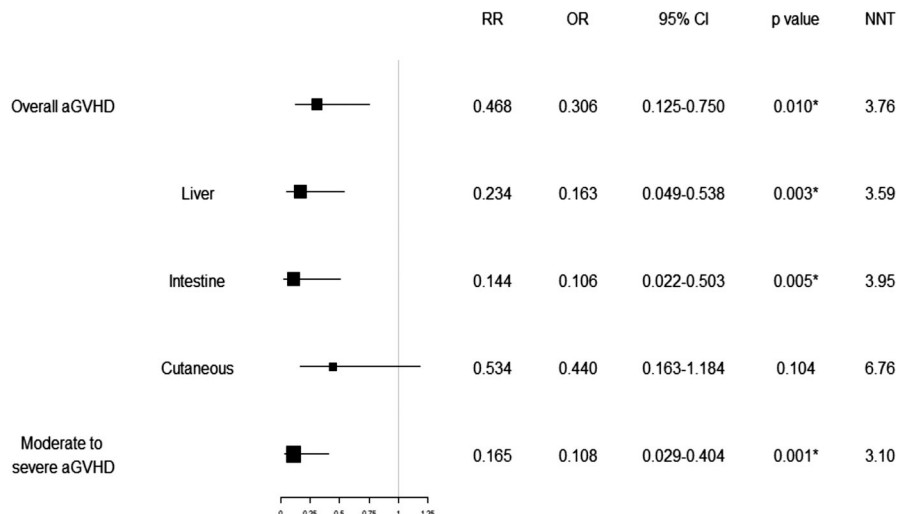

Figure 2.   Effect of defibrotide on the overall incidence and severity of acute GVHD.
The protective effect of defibrotide on the incidence of hepatic and intestinal GVHD, but not cutaneous, is shown after stratification by organ. aGVHD, acute graft-versus-host disease.

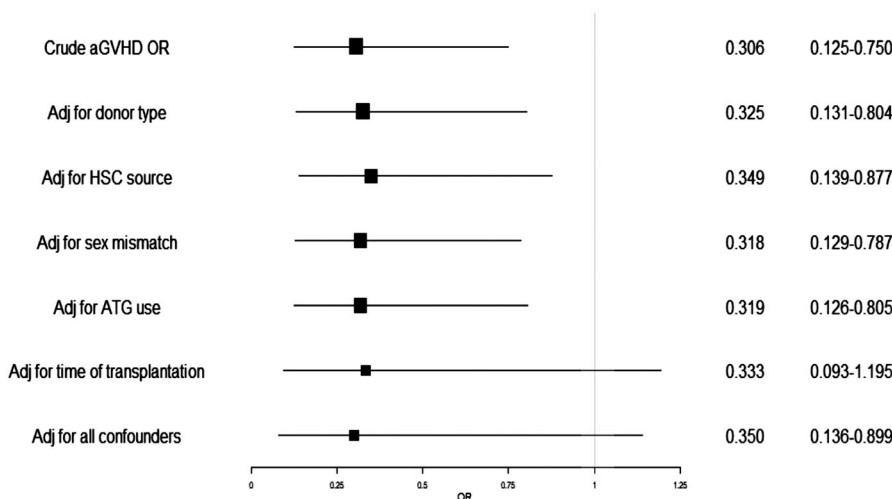

Figure 3.   Multivariate logistic regression: risk of acute GVHD adjusted for confounding factors.
aGVHD, acute graft-versus-host disease; Adj, adjusted; HSC, hematopoietic stem cell; ATG, anti-thymocyte globulin.

evidences. In particular, in vivo experiments in murine models support the involvement of IL-7 in developing GVHD ([43], [44]). High levels of IL-7 have also been observed in patients with grade II–IV aGVHD or with a higher incidence of cGVHD ([45]). This concomitance of events in patients has been associated with poor CD4[+] and CD8[+] reconstitution of T cells ([46]).

In our study, RANTES is down-modulated, suggesting its involvement in aGVHD. Nomura et al published data, where patients with aGVHD and SOS showed higher levels of this cytokine ([47]). These data were also confirmed in vivo experiments in a murine model ([48]).

On the other hand, the up-regulation of some cytokines, such as IL-1RA and IL-17, confirms that defibrotide treatment is ineffective at the skin involvement level. Several studies have shown a pleiotropic role of IL-1RA in the alloreactive response of T cells, and its inhibition is strictly correlated with the production of IFN-? and IL-17. These in vitro experiments have outlined that the inhibition of IL-1RA may effectively limit and block the progression of GVHD ([49]).

Moreover, IL-17 is a target molecule related to skin inflammation in several pathologies with alopecia. Indeed, it represents a specific target for therapeutic strategy and is also a molecule that plays an essential role in skin involvement in late-onset GVHD, like IL-6 ([50], [51]). The IL-17 is a powerful marker specific for cutaneous GVHD, and its plasma concentration increases after defibrotide treatment. This evidence should be considered to implement specific pharmacological interventions for the skin.

In contrast, IP10, another marker related to the pathogenesis of GVHD in the skin, has an opposite trend. Piper et al analyzed the expression of IP10 and its ligand in biopsies and serum of patients with GVHD. They demonstrated the correlation between high levels of IP10 and the onset of GVHD as well as the role of chemokine in the pathogenesis of skin complications associated with GVHD. Although the decreased IP10 levels in our study strengthen the role of defibrotide in preventing GVHD, it is necessary to further study the role of this cytokine in the pathogenesis of skin-associated complications ([52]).

**Table 2.  Comparison of cytokines profile between defibrotide and control groups.**

| Cytokines | Defibrotide group, median (IQR), pg/ml | Control group, median (IQR), pg/ml | P-value |
|---|---|---|---|
| IL-1b | 3.02 (1.73–5.10) | 3.76 (1.38–7.34) | 0.5295 |
| IL-1ra | 103.55 (77.24–150.55) | 25.33 (17.72–29.12) | $3.38 \times 10^{-13}$ |
| IL-2 | 9.91 (7.53–12.58) | 14.44 (10.02–16.76) | 0.001447 |
| IL-4 | 1.91 (1.24–2.96) | 9.41 (8.43–11.65) | $3.49 \times 10^{-8}$ |
| IL-5 | 0.86 (0.41–3.04) | 4.06 (1.67–7.20) | $7.04 \times 10^{-3}$ |
| IL-6 | 6.44 (4.32–14.20) | 45.49 (44.75–98.81) | $4.09 \times 10^{-10}$ |
| IL-7 | 4.84 (3.09–6.80) | 67.97 (56.49–92.04) | $<2.2 \times 10^{-16}$ |
| IL-8 | 18.54 (13.65–25.58) | 54.14 (31.06–93.08) | $3.77E \times 10^{-5}$ |
| IL-9 | 54.07 (41.16–81.42) | 43.02 (26.71–55.94) | 0.00128 |
| IL-10 | 4.39 (2.72–5.33) | 0.14 (0.14–3.07) | $7.32 \times 10^{-5}$ |
| IL-12 (p70) | 7.85 (5.92–11.03) | 9.58 (6.87–15.73) | 0.05733 |
| IL-13 | 4.47 (2.44–5.76) | 6.39 (5.07–7.59) | 0.0001542 |
| IL-15 | 10.28 (4.20–14.50) | 10.22 (8.87–14.12) | 0.3466 |
| IL-17 | 27.81 (21.20–36.22) | 6.80 (4.95–7.98) | $3.58 \times 10^{-13}$ |
| Eotaxin | 62.76 (51.17–103.09) | 117.72 (72.53–212.60) | $3.89 \times 10^{-2}$ |
| FGF basic | 25.41 (17.99–31.36) | 8.88 (7.09–10.06) | $4.21 \times 10^{-11}$ |
| G-CSF | 67.91 (52.81–99.40) | 88.29 (88.29–129.36) | 0.000489 |
| GM-CSF | 53.76 (53.75–74.21) | 51.15 (39.78–64.39) | 0.008147 |
| IFNγ | 33.39 (21.71–54.88) | 13.80 (11.92–17.36) | $3.16 \times 10^{-6}$ |
| IP-10 | 169.34 (112.65–522.84) | 1322.02 (907.90–2,074.43) | $2.61 \times 10^{-13}$ |
| MCP-1 (MCAF) | 12.71 (8.25–23.36) | 351.23 (152.75–672.62) | $<2.2 \times 10^{-16}$ |
| MIP-1 a | 5.07 (4.38–8.44) | 28.17 (14.19–42.94) | $6.15 \times 10^{-8}$ |
| PDGF-bb | 88.32 (52.52–146.51) | 34.32 (28.78–65.75) | $2.40 \times 10^{-4}$ |
| MIP-1b | 68.55 (53.56–97.39) | 193.08 (80.00–505.35) | 0.0001106 |
| RANTES | 1239.80 (983.76– 1,686.68) | 11953.13 (6953.16–17,794.79) | $9.56 \times 10^{-10}$ |
| TNFα | 35.38 (27.85–55.07) | 113.89 (95.38–128.04) | $2.33 \times 10^{-8}$ |
| VEGF | 12.46 (9.07–17.63) | 12.26 (9.42–14.83) | 0.7001 |

IQR, interquartile range.

Our study is inherently limited by its retrospective nature and the small sample size, moreover the heterogeneity of primary diseases.

In conclusion, in this study, patients receiving defibrotide prophylaxis showed a reduced incidence of aGVHD, with a modification of their cytokine pattern strongly coherent with protective drug action compared with the control group. This evidence adds to pediatric retrospective studies and preclinical data suggesting a possible defibrotide role in this setting. High costs currently limit the drug's use. However, should these data be confirmed by a large prospective pediatric trial, the positive impact on patient outcomes and the reduced long-term burdensome costs of GVHD and SOS should be considered.

# Materials and Methods

### Study design and population

This retrospective, single-center, observational study was conducted at the Pediatric Bone Marrow Transplant Center of the Institute for Maternal and Child Health–IRCCS Burlo Garofolo, Trieste, Italy, between 2010 and 2021. The IRCCS Burlo Garofolo Ethics Committee (reference no. 1105/2015) approved the protocol, and the study was conducted by the Declaration of Helsinki and Good Clinical Practice guidelines. The patients' parents gave their written consent to collect and use personal data for research purposes.

This analysis included consecutive and concurrent pediatric patients with hematological malignancies and hematological nonmalignant diseases given myeloablative conditioning followed by hematopoietic stem cell grafts from matched siblings, 10/10 matched unrelated or related haploidentical donors. We excluded the patients over 18 yr at the time of transplantation, second or subsequent transplant attempt, nonmyeloablative conditioning, mismatched unrelated donor grafts, and follow-up less than 6 mo. Fig S2 demonstrates patient eligibility and inclusion criteria. Patients included in the study were divided into two groups based on defibrotide use. All patients who received defibrotide prophylaxis were included in the defibrotide (study) group, and the patients

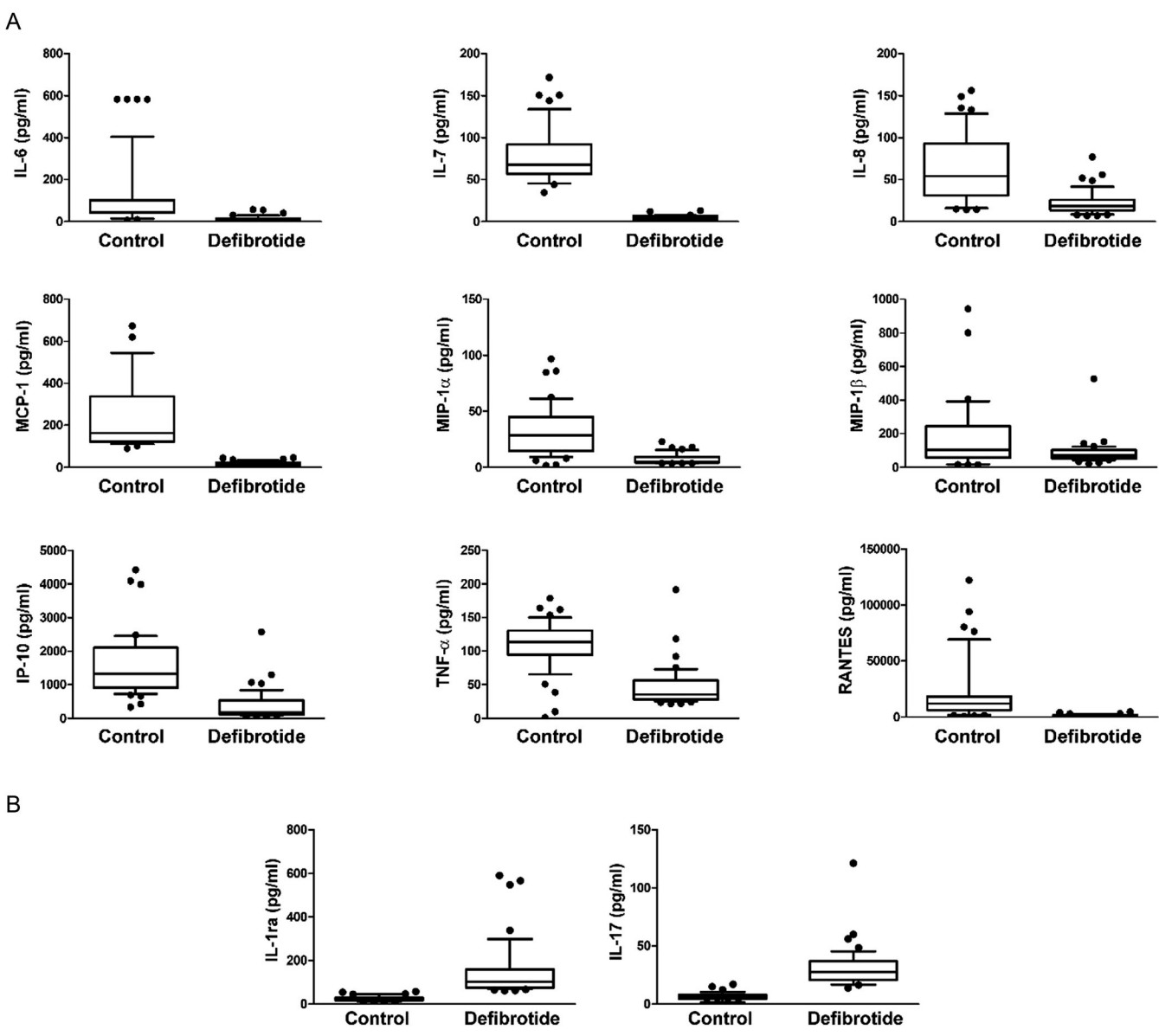

**Figure 4.   Cytokine levels in the control and defibrotide groups: IL-7, IL-6, IL8, IP-10, MCP-1, MIP-1α, MIP-1β, TNF-α, RANTES, IL-1ra, and IL-17.**
**(A, B)** Cytokines down-regulated (A) and up-regulated (B) in the defibrotide group in comparison to the control group were measured in plasma samples by multiplex immunoassays. Inside box horizontal bars are medians; the upper and lower edges of the box are 75th and 25th percentiles, respectively, and outside box horizontal bars are 10th and 90th percentiles. Defibrotide group versus control group, $P < 0.001$ in all comparisons.

who did not receive defibrotide prophylaxis were included in the control group.

## Definitions and endpoints

The disease stage at HSCT for hematological malignancies was defined according to the European Group for Blood and Marrow Transplantation (EBMT) risk score (31). The disease stage for nonmalignancies was defined as the hematopoietic cell transplant comorbidity index (HCT-CI) (32). The myeloablative conditioning regimen was defined as total body irradiation ≥ 8 Gy, busulfan

16 mg/kg, or melphalan 140 mg/m$^2$ (53). All patients were treated according to the standard myeloablative protocols based on chemotherapy and radiation dosing, as previously described (46, 54). GVHD prophylaxis was performed with tacrolimus. Additional GVHD prophylaxis included mycophenolate mofetil for the matched unrelated donor (MUD), with the addition of posttransplant cyclophosphamide from 2013 in the case of a haploidentical donor. Serotherapy with anti-thymocyte globulin was also assessed as an independent variable. Prevention and treatment of infection and other elements of transplant-specific supportive care were performed according to institutional standard practices. Duration of

follow-up was defined as the time from HSCT to last contact or death. Acute and cGVHD were diagnosed and graded using standard criteria (55, 56, 57).

The study's primary endpoints were comparing the incidence of aGVHD and chronic GVHD-free survival between the defibrotide group and the control group. The incidence of GVHD was defined as any GVHD requiring systemic immune suppressive therapy. The secondary endpoints evaluated the influence of defibrotide prophylaxis on the incidence of early and late transplant-related complications. Early transplant-related complications were defined as events occurring within 100 d after HSCT unrelated to primary disease recurrence.

### Defibrotide prophylaxis

Between January 2010 and June 2014, all patients receiving allo-HCT for hematological malignancies, autosomal recessive osteopetrosis, and thalassemia underwent SOS prophylaxis with defibrotide. From July 2014, only patients at high risk of developing SOS underwent defibrotide prophylaxis. The patient was considered at high risk of SOS developing in the presence of at least three risk factors among patient-related and transplant-related factors defined in the literature (58). Defibrotide administration started on the first day and lasted until 28 d after conditioning, at a 25 mg/kg daily dose divided into four administrations per day.

### Analysis of cytokines and chemokines

The analysis of 27 cytokines and chemokines, namely, IL-1$\beta$, IL-1ra, IL-2, IL-4, IL-5, IL-6, IL-7, IL-8, IL-9, IL-10, IL-12(p70), IL-13, IL-15, IL-17, eotaxin, FGF basic, G-CSF, GM-CSF, IFN-$\gamma$, IP-10, MCP-1 (MCAF), MIP-1$\alpha$, PDGF-bb, MIP-1$\beta$, RANTES (CCL5), TNF-$\alpha$, and VEGF was carried out on plasma samples with multiple immunoassays, using a bead-based magnetic sensor (27 human-Bio-Plex assay) (Bio-Rad Laboratories) following the manufacturer's instructions. Data were acquired by a Bio-Plex 200 reader and a digital processor, and Bio-Plex Manager 6.0 software converted data into median fluorescence intensity and concentration (pg/$\mu$l).

### Statistical analysis

Patient and transplant characteristics were expressed as the number and percentage of the group for categorical variables and median with interquartile ranges for continuous variables. We assessed the incidence rate of aGVHD in the defibrotide group and confronted it with that of the control group. Stratification was performed for moderate to severe aGVHD and organ-specific (liver, intestinal, and cutaneous) aGVHD. Multivariate analysis was performed using the logistic regression model to adjust the risk of aGVHD for possible present confounding factors: type of donor (HLA identical sibling versus MUD or haploidentical donor); sex mismatched between donor and recipient (F→M versus M→F or matched); source of hematopoietic stem cells (peripheral blood versus bone marrow); the use of anti-thymocyte globulin in GVHD prophylaxis. Chronic GVHD-free survival was calculated using the Kaplan-Meier method, and that of the two groups was confronted using log ranks. Finally, we assessed the median values of each

cytokine and chemokine and confronted that of the two groups using the non-parametrical U test method. All outcomes and variables were pre-set, and two-sided $P$-values < 0.05 were considered statistically significant. Statistical analysis was performed using R version 4.2.0.

## Supplementary Information

## Acknowledgements

This work was supported by the Ministry of Health, Rome, Italy, in collaboration with the Institute for Maternal and Child Health IRCCS Burlo Garofolo, Trieste, Italy. The authors thank Nevenka Medic for the English revision of the article.

### Author Contributions

D Squillaci: data curation and formal analysis.
A Marcuzzi: conceptualization and methodology.
E Rimondi: conceptualization and methodology.
G Riccio: data curation and formal analysis.
E Barbi: supervision and writing—review and editing.
D Zanon: validation and writing—review and editing.
N Maximova: writing—original draft and project administration.

### Conflict of Interest Statement

The authors declare that they have no conflict of interest.

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
