## [Reviewer comments · Life Science Alliance]

Life Science Alliance

Defibrotide impact on acute GVHD incidence in pediatric hematopoietic cell transplant recipients.

Domenica Squillaci, Annalisa Marcuzzi, Erika Rimondi, Guglielmo Riccio, Egidio Barbi, Davide Zanon, and Natalia Maximova
DOI: <https://doi.org/10.26508/lsa.202201786>

Corresponding author(s): Natalia Maximova, Institute for Maternal and Child Health - IRCCS Burlo Garofolo

Review Timeline:

Submission Date:	2022-10-26
Editorial Decision:	2023-01-06
Revision Received:	2023-02-23
Editorial Decision:	2023-02-24
Revision Received:	2023-02-25
Accepted:	2023-02-27

Transaction Report:

January 6, 2023

Re: Life Science Alliance manuscript #LSA-2022-01786-T

Dr. Natalia Maximova
Institute for Maternal and Child Health - IRCCS Burlo Garofolo
via dell'Istria 65/1
Trieste, Friuli-Venezia Giulia 34137
Italy

Dear Dr. Maximova,

Thank you for submitting your manuscript entitled "Defibrotide impact on the acute graft-versus-host disease incidence in pediatric hematopoietic stem cell transplant recipients" to Life Science Alliance. The manuscript was assessed by an expert reviewer, whose comments are appended to this letter. We invite you to submit a revised manuscript addressing the Reviewer comments.

When submitting the revision, please include a letter addressing the reviewer's comments point by point.

Thank you for this interesting contribution to Life Science Alliance. We are looking forward to receiving your revised manuscript.

Sincerely,

B. MANUSCRIPT ORGANIZATION AND FORMATTING:

Reviewer #1 (Comments to the Authors (Required)):

GENERAL COMMENTS :

In this study, Squillaci et al report, in a pediatric population, a lower incidence of aGVHD in patients who received defibrotide as prophylaxis for VOD. Overall, I think this study may be a good addition to the literature, it shows that defibrotide may work in the pediatric population and may explain why 2 pediatric studies showed a benefit but no adult studies did. I think that the cytokine assay reinforces the message and makes the study more robust.

But first, I think this study needs to be "polished", to properly acknowledge that defibrotide does not seem to work in the adult population and to try to explain why it seems to work in the pediatric population.

There are also some sentences that appear to have been translated "directly" from another language. I strongly suggest that you review the manuscript with a native English speaker.

MAJOR:

Introduction:

Line 81 - 84: I strongly disagree with this sentence. To date:

- The only available studies showing statistical significance of aGVHD prophylaxis with defibrotide are pediatric studies published in 2010 & 2012, evaluating defibrotide in an SOS-prophylaxis setting. These results are secondary analyses and the studies were not designed for this purpose (Richardson PG et al, doi: 10.1016/j.bbmt.2010.02.009) ; Corbacioglu S et al, doi:10.1016/S0140-6736(11)61938-7)
- Another study investigating defibrotide prophylaxis in an adult population showed a trend in favor of defibrotide as aGVHD prophylaxis but did not reach statistical significance (Tekgündüz E et al, doi:10.1016/j.transci.2016.01.009.)
- More recently, a retrospective case-control study in an adult population found no benefit for defibrotide in aGVHD prophylaxis but did find worse overall survival in the defibrotide group (Tilmont R et al doi:10.1177/10600280211068177)
- Finally, a recent phase-2, prospective, randomized clinical trial investigating the efficacy of defibrotide prophylaxis added to standard of care versus standard of care alone for aGVHD did not find a benefit, both in aGVHD incidence or survival. The study is not published yet but you can find the raw results at clinicaltrials.gov (<https://clinicaltrials.gov/ct2/show/results/NCT03339297?view=results>)

I would modify this sentence to reflect the fact that this issue is being debated, and I would strongly suggest that the relevant section be modified in the discussion (see below). Furthermore, as this is a controversial and long debated subject, I think that the list of references should be exhaustive.

Line 87: This section also lacks explanation on defibrotide. I would move the first part of the discussion (line 132-139) here.

Results:

Line 97 - 98: There is also more "bone marrow" in the defibrotide group. This is a very important confounding factor, as "peripheral blood" transplants contain more T cells and are generally associated with a higher risk of aGVHD. This parameter is already included in the multivariate analysis, but you should mention it here and acknowledge it in the caveats as well.

Line 130: In this section, data on overall survival and aGVHD-free survival with relevant figures are definitely missing.

Discussion:

Line 132 -139: I would move this part to the Introduction.

Line 148 - 164: As mentioned above, I strongly disagree with this part because most studies of defibrotide in aGVHD have been non-significant and/or negative. In addition, the EMA no longer recommends the use of defibrotide as prophylaxis for VOD. (<https://www.ema.europa.eu/en/medicines/dhpc/defitelio-defibrotide-do-not-use-prophylaxis-veno-occlusive-disease-vod-after-post-hematopoietic>).

In this case, I would strongly suggest pointing out that these results contradict the latest studies. I would also add a sentence or two to try to explain why this seems to work in a pediatric population, but not in adults. Perhaps a difference in cytokine balance?

Line 214-221: I would change this part, as per my previous comments.

Methods:

Line 226: This study included patients from 2010 to 2021. Were there any differences in patient management, including aGVHD prophylaxis? I would also add timing of transplantation as a cofactor in the multivariate analysis. Before and after 2014 seems to be a good compromise because the indication for defibrotide prophylaxis has changed.

Line 269: What reference did you follow on the duration of defibrotide prophylaxis ? What was your policy if the patient achieved engraftment before the end of the defibrotide prophylaxis ?

The average effective duration of defibrotide prophylaxis in the population should also appear in the Results. It might also be interesting to see if patients who received a higher total dose of defibrotide had better protection against aGVHD and/or higher overall survival.

MINOR:

Abstract:

Line 8: Typo, « aGV6hD »

Introduction:

Line 52: I strongly suggest using allo-HCT rather than alloHSCT. This is now the appropriate terminology (and you mention the "HCT Comorbidity Index" in the Methods 244 line).

Line 57: As you write later about mild and severe aGVHD, I would add a sentence about the severity scale of aGVHD.

Results:

Line 102: Typo, -a

Line 119: "We were not found" has no meaning in English.

Line 169: Typo, "Of note, the comparison between"

Line 238: Typo, "and the patients who non received defibrotide"

GENERAL COMMENTS :

In this study, Squillaci et al report, in a pediatric population, a lower incidence of aGVHD in patients who received defibrotide as prophylaxis for VOD. Overall , I think this study may be a good addition to the literature, it shows that defibrotide may work in the pediatric population and may explain why 2 pediatric studies showed a benefit but no adult studies did. I think that the cytokine assay reinforces the message and makes the study more robust.

Thank you.

But first, I think this study needs to be "polished", to properly acknowledge that defibrotide does not seem to work in the adult population and to try to explain why it seems to work in the pediatric population.

We explained this question.

There are also some sentences that appear to have been translated "directly" from another language. I strongly suggest that you review the manuscript with a native English speaker.

The manuscript has been undergoing English editing.

MAJOR:

Introduction:

Line 81 - 84: I strongly disagree with this sentence. To date:

- The only available studies showing statistical significance of aGVHD prophylaxis with defibrotide are pediatric studies published in 2010 & 2012, evaluating defibrotide in an SOS-prophylaxis setting. These results are secondary analyses and the studies were not designed for this purpose (Richardson PG et al, doi: 10.1016/j.bbmt.2010.02.009) ; Corbacioglu S et al, doi:10.1016/S0140-6736(11)61938-7)
- Another study investigating defibrotide prophylaxis in an adult population showed a trend in favor of defibrotide as aGVHD prophylaxis but did not reach statistical significance (Tekgündüz E et al, doi:10.1016/j.transci.2016.01.009.)

- More recently, a retrospective case-control study in an adult population found no benefit for defibrotide in aGVHD prophylaxis but did find worse overall survival in the defibrotide group (Tilmont R et al doi:10.1177/10600280211068177)
- Finally, a recent phase-2, prospective, randomized clinical trial investigating the efficacy of defibrotide prophylaxis added to standard of care versus standard of care alone for aGVHD did not find a benefit, both in aGVHD incidence or survival. The study is not published yet but you can find the raw results at [clinicaltrials.gov](https://clinicaltrials.gov/ct2/show/results/NCT03339297?view=results) (<https://clinicaltrials.gov/ct2/show/results/NCT03339297?view=results>)

I would modify this sentence to reflect the fact that this issue is being debated, and I would strongly suggest that the relevant section be modified in the discussion (see below). Furthermore, as this is a controversial and long debated subject, I think that the list of references should be exhaustive.

Thank you for your advice. We modified the sentence and added a complete list of references on this issue (lines 89 – 91).

Line 87: This section also lacks explanation on defibrotide. I would move the first part of the discussion (line 132-139) here.

We have moved this paragraph to the Introduction section (lines 81 – 88).

Results:

Line 97 - 98: There is also more "bone marrow" in the defibrotide group. This is a very important confounding factor, as "peripheral blood" transplants contain more T cells and are generally associated with a higher risk of aGVHD.

Yes, this statement is correct for the adult population.

Two recent studies demonstrated no differences in the incidence of acute GVHD of any grade in HSCT with MMUD + ATG and haploidentical donor + PTCy in pediatric patients.

*Berger M, Barone M, Spadea M, Saglio F, Pessolano R, Fagioli F. HSCT with mismatched unrelated donors: Bone marrow versus peripheral blood stem cells sources in pediatric patients. *Pediatr Transplant.* 2022 Jun;26(4):e14233. doi: 10.1111/ptr.14233.*

*Srinivasan A, Raffa E, Wall DA, Schechter T, Ali M, Chopra Y, Kung R, Chiang KY, Krueger J. Outcome of Haploidentical Peripheral Blood Allografts Using Post-Transplantation Cyclophosphamide Compared to Matched Sibling and Unrelated Donor Bone Marrow Allografts in Pediatric Patients with Hematologic Malignancies: A Single-Center Analysis. *Transplant Cell Ther.* 2022 Mar;28(3):158.e1-158.e9. doi: 10.1016/j.jtct.2021.11.009.*

In our study population, the prevalence of PBSC as a graft source is associated with more frequent use of ATG and the greater number of haploidentical transplants with post-transplant cyclophosphamide. Logistic regression showed that the protective effect of defibrotide prophylaxis on aGVHD is maintained after adjustment for graft source (Figure 3).

Line 130: In this section, data on overall survival and aGVHD-free survival with relevant figures are definitely missing.

In the Results section, we added the sentence describing OS and aGVHD-free survival (lines 110-111 and 117 - 118). Also, we added Figure 1A (overall survival) and Figure 1B (aGVHD-free

survival).

Discussion:

Line 132 -139: I would move this part to the Introduction.

We have moved this paragraph to the Introduction section.

Line 148 - 164: As mentioned above, I strongly disagree with this part because most studies of defibrotide in aGVHD have been non-significant and/or negative. In addition, the EMA no longer recommends the use of defibrotide as prophylaxis for VOD.

<https://www.ema.europa.eu/en/medicines/dhpc/defitelio-defibrotide-do-not-use-prophylaxis-veno-occlusive-disease-vod-after-post-hematopoietic>.

In this case, I would strongly suggest pointing out that these results contradict the latest studies. I would also add a sentence or two to try to explain why this seems to work in a pediatric population, but not in adults. Perhaps a difference in cytokine balance?

We have modified this paragraph as suggested (lines 152 – 174 and 181 – 186).).

Line 214-221: I would change this part, as per my previous comments.

We have modified this part as per your suggestions (lines 235 – 241).

Methods:

Line 226: This study included patients from 2010 to 2021. Were there any differences in patient management, including aGVHD prophylaxis? I would also add timing of transplantation as a cofactor in the multivariate analysis. Before and after 2014 seems to be a good compromise because the indication for defibrotide prophylaxis has changed.

We added the timing of transplantation as a cofactor in the multivariate analysis (Figure 3).

Line 269: What reference did you follow on the duration of defibrotide prophylaxis? What was your policy if the patient achieved engraftment before the end of the defibrotide prophylaxis? The average effective duration of defibrotide prophylaxis in the population should also appear in the Results. It might also be interesting to see if patients who received a higher total dose of defibrotide had better protection against aGVHD and/or higher overall survival.

Dosage, method of administration, and duration of prophylaxis with defibrotide were defined by the study protocol proposed by the Gentium company, which supplied the drug for the study. We maintained the same defibrotide administration protocol after the study closure in 2014.

Therefore, all patients received the same dose of defibrotide per kg and for the same time frame.

Duration of prophylaxis was not modified based on engraftment; on day +28, almost all patients usually obtain engraftment.

In the Results section, we added the sentence about the mean duration of defibrotide prophylaxis (lines 107-109).

MINOR:

Abstract:

Line 8: Typo, « aGV6hD »

We corrected it.

Introduction:

Line 52: I strongly suggest using allo-HCT rather than alloHSCT. This is now the appropriate terminology (and you mention the "HCT Comorbidity Index" in the Methods 244 line).

We replaced the term alloHSCT with allo-HCT.

Line 57: As you write later about mild and severe aGVHD, I would add a sentence about the severity scale of aGVHD.

The sentence about the severity scale is found in the Methods section Definitions and Endpoints (line 274).

Results:

Line 102: Typo, -a

We deleted "a".

Line 119: "We were not found" has no meaning in English.

We corrected the sentence.

Line 169: Typo, "Of note, the comparison between"

We deleted this typo.

Line 238: Typo, "and the patients who non received defibrotide"

We corrected it.

February 24, 2023

RE: Life Science Alliance Manuscript #LSA-2022-01786-TR

Dr. Natalia Maximova
Institute for Maternal and Child Health - IRCCS Burlo Garofolo
via dell'Istria 65/1
Trieste, Friuli-Venezia Giulia 34137
Italy

Dear Dr. Maximova,

Thank you for submitting your revised manuscript entitled "Defibrotide impact on acute GVHD incidence in pediatric hematopoietic cell transplant recipients.". We would be happy to publish your paper in Life Science Alliance pending final revisions necessary to meet our formatting guidelines.

- please upload both your main and supplementary figures as single files
- please add an abstract, summary blurb, and a category to our system
- please add the Twitter handle of your host institute/organization as well as your own or/and one of the authors in our system
- please use the [10 author names, et al.] format in your references (i.e. limit the author names to the first 10)
- please upload your table files as editable doc or excel files
- Clinical trials should be registered and the trial registration number should be provided in the Materials and Methods

A. FINAL FILES:

B. MANUSCRIPT ORGANIZATION AND FORMATTING:

**Submission of a paper that does not conform to Life Science Alliance guidelines will delay the acceptance of your

manuscript.**

The license to publish form must be signed before your manuscript can be sent to production. A link to the electronic license to publish form will be sent to the corresponding author only. Please take a moment to check your funder requirements.

Sincerely,

February 27, 2023

RE: Life Science Alliance Manuscript #LSA-2022-01786-TRR

Dr. Natalia Maximova
Institute for Maternal and Child Health - IRCCS Burlo Garofolo
via dell'Istria 65/1
Trieste, Friuli-Venezia Giulia 34137
Italy

Dear Dr. Maximova,

Thank you for submitting your Research Article entitled "Defibrotide impact on acute GVHD incidence in pediatric hematopoietic cell transplant recipients.". It is a pleasure to let you know that your manuscript is now accepted for publication in Life Science Alliance. Congratulations on this interesting work.

DISTRIBUTION OF MATERIALS:

Again, congratulations on a very nice paper. I hope you found the review process to be constructive and are pleased with how the manuscript was handled editorially. We look forward to future exciting submissions from your lab.

Sincerely,
